# The Diagnosis of Autoimmune Pancreatitis Using Endoscopic Ultrasonography

**DOI:** 10.3390/diagnostics10121005

**Published:** 2020-11-25

**Authors:** Atsushi Kanno, Eriko Ikeda, Kozue Ando, Hiroki Nagai, Tetsuro Miwata, Yuki Kawasaki, Yamato Tada, Kensuke Yokoyama, Norikatsu Numao, Jun Ushio, Kiichi Tamada, Alan Kawarai Lefor, Hironori Yamamoto

**Affiliations:** 1Department of Medicine, Division of Gastroenterology, Jichi Medical University, Shimotsuke 329-0498, Japan; r1403ie@jichi.ac.jp (E.I.); kozue_ando@jichi.ac.jp (K.A.); m05069hn@jichi.ac.jp (H.N.); tetsurou_miwata@yahoo.ac.jp (T.M.); kawasakiyuki1219@gmail.com (Y.K.); tadayamatoday@gmail.com (Y.T.); r0760ky@jichi.ac.jp (K.Y.); numawo@jichi.ac.jp (N.N.); j.ushio@jichi.ac.jp (J.U.); tamadaki@jichi.ac.jp (K.T.); yamamoto@jichi.ac.jp (H.Y.); 2Department of Surgery, Jichi Medical University, Shimotsuke 329-0498, Japan; alefor@jichi.ac.jp

**Keywords:** autoimmune pancreatitis (AIP), endoscopic ultrasound (EUS), EUS-guided fine needle aspiration (EUS-FNA), EUS-guided fine needle biopsy (EUS-FNB), International Consensus Diagnostic Criteria (ICDC)

## Abstract

Autoimmune pancreatitis (AIP) is characterized by enlargement of the pancreas and irregular narrowing of the main pancreatic duct. It is often associated with IgG4-related sclerosing cholangitis (IgG4-SC), in which the bile duct narrows. Although characteristic irregular narrowing of the pancreatic duct caused by endoscopic retrograde cholangiopancreatography is noted in AIP, it is difficult to differentiate between localized AIP and pancreatic carcinoma based on imaging of the pancreatic duct. While stenosis of the bile duct in IgG4-SC is characterized by longer-length stenosis than in cholangiocarcinoma, differentiation based on bile duct imaging alone is challenging. Endoscopic ultrasound (EUS) can characterize hypoechoic enlargement of the pancreas or bile duct wall thickening in AIP and IgG4-SC, and diagnosis using elastography and contrast-enhanced EUS are being evaluated. The utility of EUS-guided fine needle aspiration for the histological diagnosis of AIP has been reported and is expected to improve diagnostic performance for AIP. Findings in the bile duct wall from endoscopic retrograde cholangiopancreatography followed by intraductal ultrasonography are useful in differentiating IgG4-SC from cholangiocarcinoma. Diagnoses based on endoscopic ultrasonography play a central role in the diagnosis of AIP.

## 1. Introduction

The concept of autoimmune pancreatitis (AIP) was first proposed by Yoshida et al. in 1995 [1]. Subsequently, there were reports about the diagnosis and treatment of AIP. Many patients with AIP present with other organ involvement, such as sclerosing cholangitis and sialadenitis, and AIP is considered a pancreatic lesion of systemic IgG4-related disease (IgG4-RD) [2]. Diagnostic criteria have been reported from various countries. The diagnostic criteria in Japan were released in 2002 [3] and subsequently revised three times [4,5]. The International Consensus Diagnostic Criteria (ICDC) for AIP [6] were published in 2011 based on the Honolulu consensus [7]. Due to the process by which the current diagnostic criteria were developed, the importance of endoscopic ultrasound (EUS) in the diagnosis of AIP needs to be clarified. This review shows the role and progress of EUS in the diagnosis of AIP.

## 2. Diagnostic Criteria for AIP

In 2009, a joint meeting between the American Pancreatic Association and the Japan Pancreas Society (JPS) was held in Honolulu to discuss about the concept of AIP [7]. This meeting resulted in the release of the Honolulu consensus, which emphasized the histological findings in types 1 and 2 AIP showing lymphoplasmacytic sclerosing pancreatitis and idiopathic duct-centric chronic pancreatitis/granulocytic epithelia lesions, respectively [7]. Based on the Honolulu consensus, ICDC for AIP [6] were published after further discussion at an international joint meeting between the International Pancreas Society and JPS held in Fukuoka, Japan in July 2010. In the ICDC, AIP is diagnosed based on the presence/absence of a combination of five findings: (i) pancreatic imaging findings (parenchyma (P) and pancreatic duct (D)), (ii) serologic findings (S), (iii) other organ involvement (O), (iv) histological findings (H), and (v) response to steroid therapy (Rt). The ICDC enables the diagnosis of AIP using global diagnostic criteria. However, the ICDC were overly complicated for general clinicians to diagnose patients with AIP and Type 2 AIP is extremely rare in Japan. Therefore, the JPS developed Clinical Diagnostic Criteria for AIP 2011 [5] (JPS 2011) based on the ICDC. Later, these criteria were further revised as the Clinical Diagnostic Criteria for AIP 2018 (JPS 2018) [8]. Currently, the diagnosis of AIP by EUS plays an important role in the ICDC of JPS 2018.

## 3. The Role of Endoscopy in the Diagnosis of AIP

### 3.1. Endoscopic Retrograde Cholangiopancreatography

While this review focuses on EUS, endoscopic retrograde cholangiopancreatography (ERCP) will be briefly described because ERCP is also an essential modality for the diagnosis of AIP and is an essential procedure for intraductal ultrasonography (IDUS). Pancreatic enlargement with diffuse irregular narrowing of the main pancreatic duct (MPD) is a typical imaging finding in patients with AIP [1]. Narrowing of the MPD is defined as a finding where smaller than normal pancreatic ducts dilate and present with irregular margins (Figure 1) [9]. ERCP is essential for diagnosing irregular narrowing of the MPD. Level 1 narrowing of the MPD was defined as at least one-third of the total length of the pancreatic duct without a dilatation of pancreatic body and tail in the ICDC [6]. The ICDC [6] and JPS2011 [5] mention narrowing of the MPD in ERCP as a condition for the definitive diagnosis of localized AIP. However, differentiating stenosis of the MPD due to pancreatic cancer from narrowing of the MPD in localized AIP is extremely difficult.

There are numerous reports of ERCP findings observed in AIP and pancreatic cancer [9,10,11]. Although the narrowing of at least one-third of the entire length of the MPD has been shown to be an important characteristic of pancreatic duct images in AIP, there have been reports of the sensitivity, specificity, and interobserver agreement of diagnosis of AIP using ERCP as being low [11]. In addition, it is important to make a comprehensive diagnosis along with other findings because there is a risk of post-ERCP pancreatitis.

AIP is sometimes complicated by IgG4-related sclerosing cholangitis (IgG4-SC) [12,13]. However, IgG4-SC is difficult to differentiate from primary sclerosing cholangitis and malignant tumors [14,15]. Most patients with primary sclerosing cholangitis have progressive disease and many develop stenosis in both the intrahepatic and extrahepatic bile ducts, leading to liver cirrhosis. In contrast, IgG4-SC has a good clinical course with steroid treatment. Obviously, it is important to distinguish between IgG4-SC and bile duct cancer. In the differentiating IgG4-SC and cholangiocarcinoma, the presence or absence of serum IgG4 or IgG4-RD should be comprehensively assessed by referring to the pathological findings of the stenotic region of the bile duct. Since it is difficult to make the definitive diagnosis of IgG4-SC based on pathological findings from a bile duct biopsy alone [16,17,18,19], the findings of liver [19,20,21] and duodenal papilla biopsies [22] are useful to establish the diagnosis. During ERCP, transpapillary IDUS provides high-resolution images of the bile duct wall, making it useful in the evaluation of bile duct wall thickening [17,23,24].

However, as we mentioned above, ERCP and IDUS have a risk of post-ERCP pancreatitis. To avoid these risks, magnetic resonance imaging (MRI) and magnetic resonance cholangiopancreatography (MRCP) should be considered for the diagnosis of AIP [25,26,27] because there are no risks of post-ERCP pancreatitis or allergies to contrast media. Furthermore, MRCP has the benefit of showing the MPD in the pancreatic tail distal to a stenosis of the pancreatic duct and the pancreatic parenchyma.

### 3.2. IDUS

During ERCP, transpapillary IDUS yields high-resolution images of the bile duct wall, making it useful to evaluate bile duct wall thickening [23,24]. IDUS findings for IgG4-SC are characterized by round symmetric wall thickening, smooth outer and inner layers, and homogeneous internal echo findings at the site of biliary stenosis. IDUS findings in cholangiocarcinoma include asymmetric wall thickening, concave outer layers, hard inner papillary layers, and heterogeneous internal echographic findings at the site of stenosis. Naitoh et al. [17] reported that wall thickening (cutoff value: 0.8 mm) of the nonstenotic region in the cholangiogram was useful for differentiating IgG4-SC from cholangiocarcinoma. However, Kuwatani et al. stated that IDUS images alone are not sufficient to differentiate between IgG4-SC and cholangiocarcinoma and that serum IgG and IgG4 levels should also be included in the diagnosis [28]. Physicians need to recognize that IDUS is simply one finding to establish the diagnosis of IgG4-SC and that other findings need to be combined for a comprehensive diagnosis.

### 3.3. EUS

#### 3.3.1. Conventional EUS

Differentiating AIP and pancreatic cancer based on hypoechoic masses using conventional EUS [29,30,31,32] is difficult (Figure 2a). EUS findings of IgG4-SC present as bile duct wall thickening (Figure 2b) and differentiating it from cholangiocarcinoma and primary sclerosing cholangitis is another matter. Hoki et al. [30] reported that conventional EUS revealed findings of diffuse hypoechoic enlargement, bile duct wall thickening, and surrounding hypoechoic zones in AIP compared to findings in pancreatic cancer. AIP is known to often present with chronic pancreatitis-like findings such as hyperechoic foci or hyperechoic strands; Okabe et al. [32] reported that these findings persisted despite treatment with steroids. We should recognize that AIP presents with different ultrasound images depending on the stage.

#### 3.3.2. Contrast-Enhanced EUS

##### Ultrasound Contrast Agents/Contrast-Enhanced Ultrasound

Following intravenous administration, ultrasound contrast agents must pass through pulmonary blood vessels and reach the periphery without the bubbles collapsing. Sonazoid^®^ is a second-generation ultrasound contrast agent that is approximately 3 μm in size and has a phospholipid membrane covering perflubutane [33,34]. The ability to be taken up by Kupffer cells has been used to identify liver tumors, as well as to image nonlinear signals generated by low-pressure ultrasound and resonance of the contrast agent, enabling the acquisition of peripheral angiography and perfusion images of parenchymal organs. Thus, Sonazoid^®^ is anticipated to be utilized in contrast-enhanced ultrasound of abdominal parenchymal organs such as the pancreas. Recently, the utility of contrast-enhanced ultrasonography for the differential diagnosis of pancreatic diseases has been reported [35,36,37,38,39,40,41]. Kitano et al. reported that contrast-enhanced ultrasonography enhances the clarity of images of pancreatic tumors allowing differentiation of the contrast pattern, which was useful for the differential diagnosis of such tumors [42]. Faccioli et al. reported that contrast-enhanced ultrasonography clarified the margins of pancreatic tumors and was useful for determining eligibility for surgical resection [36]. Other studies have used contrast-enhanced ultrasound to assess pancreatic viability prior to transplantation.

##### Contrast-Enhanced Doppler EUS

Digital EUS is available and enables rendering of Doppler images. Using ultrasound contrast agents for Doppler imaging enhances the signal, making it possible to obtain a clearer image of blood flow. Doppler signals exhibit blooming artifacts that may interfere with imaging. However, the eFLOW mode of Aloka α10, the H-FLOW mode of Olympus ME2, and the F-FLOW mode of Fujifilm suppress blooming artifacts, producing clear images of blood flow, making them suitable for contrast-enhanced Doppler EUS. A small number of reports have described pancreatic tumors diagnosed using contrast-enhanced Doppler EUS. Dietrich et al. reported that contrast-enhanced EUS using the Doppler method was performed on 93 patients with pancreatic tumors, with the pancreatic cancer being rendered as hypovascular with excellent diagnostic capability [43]. Hocke et al. reported on a patient who was diagnosed with AIP via contrast-enhanced EUS using the Doppler method [44]. The Doppler signal enhancement effect of ultrasound contrast agents is useful for determining the presence or absence of blood flow.

##### Contrast-Enhanced Harmonic EUS

When low-pressure ultrasound is applied to an intravascular ultrasound contrast agent, the bubble diameter changes with the period of the ultrasound waves, generating harmonics. Contrast-enhanced harmonic EUS (CEH-EUS) selectively renders the second harmonic generated from an ultrasound contrast agent, enabling the acquisition of not only capillary but also parenchymal perfusion images (Figure 3).

CEH-EUS has not only enabled the acquisition of clear vascular images but also the drawing of time intensity curves and the graphic representation of changes in brightness values over time via imaging [45,46,47]. Imazu et al. [47] reported that time intensity curves using CEH-EUS were useful for differentiating AIP from pancreatic cancer.

#### 3.3.3. Elastography

Mei et al. [48] reported that the sensitivity, specificity, and odds ratio of elastography in differentiating benign from malignant solid pancreatic masses were 0.95 (95% confidence intervals (CI): 0.94–0.97), 0.67 (95% CI: 0.61–0.73), and 42.28 (95% CI: 26.90–66.46), respectively, based on a meta-analysis. Dietrich et al. [49] evaluated the utility of elastography in the diagnosis of AIP and reported characteristic patterns of elastography not only at the site of AIP masses but also in the surrounding pancreatic tissue. While elastography is expected to be used as a diagnostic method for pancreatic masses, further improvement is required because, in the case of EUS, there are problems with accuracy, such as the need to rely on the heartbeat to compress the ultrasound probe.

#### 3.3.4. EUS-Fine Needle Aspiration

The ICDC highlights the importance of histological diagnoses in the diagnosis of AIP [6]. According to the ICDC, only tissue specimens obtained by core biopsy or resection are suitable for histopathological diagnoses of AIP [6,50]. There is no clear definition of the term core biopsy in the ICDC. In the literature, it refers to a specimen taken using EUS-guided Trucut needle biopsies [50], and specimens obtained via EUS-guided fine needle aspiration (EUS-FNA) were difficult to clearly specify as core biopsies even if sufficient samples were obtained. However, the utility of EUS-FNA in the histologic diagnosis of AIP was reported in numerous patients. EUS-FNA using a 19-gauge (G) needle is also reportedly useful [51]. However, the procedure is technically difficult and there are concerns regarding complications caused by thick puncture needles [52]. Recently, the utility of 22G needles in EUS-FNA for the histological diagnosis of AIP has been reported [53,54,55,56]. Ishikawa et al. [53] reported the diagnostic utility of EUS-FNA using a 22G needle for Type 1 and Type 2 AIP—in particular, IgG4-negative AIP. Kanno et al. [54] reported that AIP could be histologically diagnosed in 20 out of 25 patients (80%) based on the ICDC. Morishima et al. [55] and Kanno et al. [56] established prospective histological diagnoses of AIP using EUS-FNA at multiple institutions and were able to demonstrate its utility (Figure 4 and Figure 5). While Kanno et al. [56] reported that histological examination of AIP reached a diagnosis in all cases, those cases where sufficient histological material is obtained using EUS-FNA could be diagnosed with high accuracy, so acquiring sufficient histological material is important.

The widespread use of EUS-FNA and the development of various EUS-FNA needles [57,58] (Figure 6a) have improved the quality and quantity of histological specimens. There are reports of histological evaluation of AIP specimens obtained using the recently developed EUS-guided fine needle biopsy (FNB) needles (Figure 6b,c) [59,60,61,62]. A systematic review to compare the tissue acquisition of pancreatic tissue in patients with AIP with EUS-FNA and EUS-FNB needles revealed that the diagnostic yield might be better with the FNB needle than with FNA needles [63]. In the future, histopathological examination using EUS may play a central role in the diagnosis of AIP.

## 4. EUS-FNA Procedures

Since obtaining sufficient histological material using EUS-FNA is particularly important to diagnose AIP correctly, we should understand the various FNA procedures.

### 4.1. Puncture Needles

Various puncture needles are commercially available. In terms of size, 19G, 20G, 22G, and 25G needles are available. In a meta-analysis comparing the diagnostic performance of 22G and 25G needles for pancreatic tumors, the pooled sensitivities of 22G and 25G needles were 0.85 and 0.93, respectively, demonstrating the superiority of 25G needles [64]. Conversely, in a randomized controlled study comparing 19G and 22G needles for the diagnosis of pancreatic tumors, the success rate of EUS-FNA was significantly lower for lesions in the pancreatic head obtained with a 19G needle than with a 22G needle. However, the overall diagnostic performance tended to be higher using a 19G needle than with a 22G needle [65]. Recently, the availability of EUS-FNB needles has expanded options. Franseen-like (Acquire; Boston Scientific Corp, Natick, MA, USA) needles are manipulated so that tissue is punctured and grasped with three tips, and specimens are collected by cutting out the tissue with three cutting planes. For this reason, a sample sufficient for accurate tissue diagnosis can be collected. Side-fenestrated needles (ProCore; Cook Medical, Bloomington, Indiana, USA) are available to acquire histologic sample. EUS-FNA needs to be performed while being aware that the diagnostic performance differs depending on the diameter and shape of the puncture needle and the puncture site.

### 4.2. Puncture Technique

Various techniques for puncturing the sites of lesions have been reported. These include the “door-knocking” method, in which a sample of tissue is pulled into the lumen of a needle by firmly pushing the needle through, and the Fanning method [66], in which tissue is are extracted at various sites by moving the needle in a fan shape with a forceps bending device. The “door-knocking” procedure is particularly important for adequate tissue collection.

In the past, generally speaking, suction was applied with an approximately 10–20 mL syringe. However, to prevent contamination with blood clots, the “without stylet” method [67], in which negative pressure is not applied has been reported. Various improvements have been developed, such as the “slow-pull” method [68], in which low negative pressure is applied by pulling the stylet slowly, the “wet-suction” method [69], in which negative pressure is applied after the puncture needle lumen is filled with saline, and the “high negative pressure” method [70], in which high suction pressure is applied using a balloon inflator. There is also a need to change the suction method depending on the rigidity of the tumor to be punctured.

### 4.3. Processing Pathological Specimens

The processing of pathological specimens is important to improve the accuracy of EUS-FNA. Rapid onsite examination performed by a cytopathologist improves the rate of pathological diagnosis using EUS-FNA. According to a report by Iglesias-Garcia et al., the accuracy of EUS-FNA in diagnosing pancreatic tumors without performing rapid onsite examination was 86.2%, while rapid onsite examination improved the histopathological accuracy to 96.8% [71]. Iwashita et al. reported the utility of collecting sufficient white tissue samples in samples collected via EUS-FNA (macroscopic onsite quality evaluation) [72]. Confirming the quality of the specimen immediately after puncture is important for securing an adequate sample to enable a pathological diagnosis.

## 5. Can Endoscopic Diagnosis Rule out Pancreatic Cancer or Cholangiocarcinoma?

In order to differentiate between AIP and IgG4-SC and malignant disease, the ICDC [6] mention “Pancreatic cancer ruled out by EUS-FNA” in the marginal notes, and the Clinical Diagnostic Criteria for IgG4-SC 2012 [13] state “Malignancies such as cholangiocarcinoma and pancreatic cancer need to be ruled out” in the lower column. Can pancreatic cancer and cholangiocarcinoma really be completely ruled out?

The histopathological diagnosis of pancreatic cancer using specimens obtained with EUS-FNA has an extremely high accuracy. Chen et al. performed a meta-analysis of the histopathological diagnostic ability of EUS-FNA limited to pancreatic cancer and reported an extremely high diagnostic ability, with a pooled sensitivity of 0.89 (95% CI: 0.88–0.90) and a pooled specificity of 0.96 (95% CI: 0.95–0.97) [73]. Meta-analyses of pancreatic tumors also exhibited good diagnostic performance in general [74,75], and it is well established that EUS-FNA is an important modality in the histopathologic diagnosis of pancreatic cancer. However, the accuracy is not 100%. Crinò et al. reported that the size of the pancreatic mass affects the accuracy of diagnosis by EUS-FNA of solid pancreatic lesions. We should be aware of the limitations of the histological diagnostic ability of EUS-FNA [76].

A meta-analysis evaluating contrast-enhanced EUS for the diagnosis of pancreatic masses revealed that the sensitivity and specificity of CE-EUS to differentiate pancreatic adenocarcinoma from other pancreatic masses was 94% and 89%, respectively [77]. In the cases in which is difficult to perform EUS-FNA, contrast-enhanced EUS is a useful imaging modality to differentiate lesions from pancreatic cancer.

To establish the histopathological diagnosis of bile duct lesions, the transpapillary approach followed by ERCP is the standard, even with a risk of dissemination with the tissue sampling method using EUS-FNA [78] and the percutaneous transhepatic approach [79]. However, the diagnostic accuracy of benign/malignant diagnoses from the main tumor site is not particularly high, at approximately 56–86% [80,81,82,83,84,85]. Thus, pancreatic biopsies using EUS-FNA and bile duct biopsies using the transpapillary approach cannot completely rule out malignancy. Although EUS-FNA and transpapillary bile duct biopsies are essential in the diagnostic process, the diagnosis of AIP and IgG4-SC should be made based on an understanding of the present limitations of pathological diagnoses using endoscopy.

## 6. Conclusions

The diagnosis of AIP using endoscopy was reviewed with a particular focus on endoscopic ultrasonography. AIP and IgG4-SC must be differentiated from malignant tumors, and ERCP and EUS play a pivotal role in establishing the diagnosis. However, there is a need to carefully diagnose AIP in conjunction with serological findings and diagnoses of other organ involvement, keeping in mind the limitations of endoscopic diagnosis. It is anticipated that AIP will be diagnosed more accurately in the future as new approaches are developed.

## Figures and Tables

**Figure 1 diagnostics-10-01005-f001:**
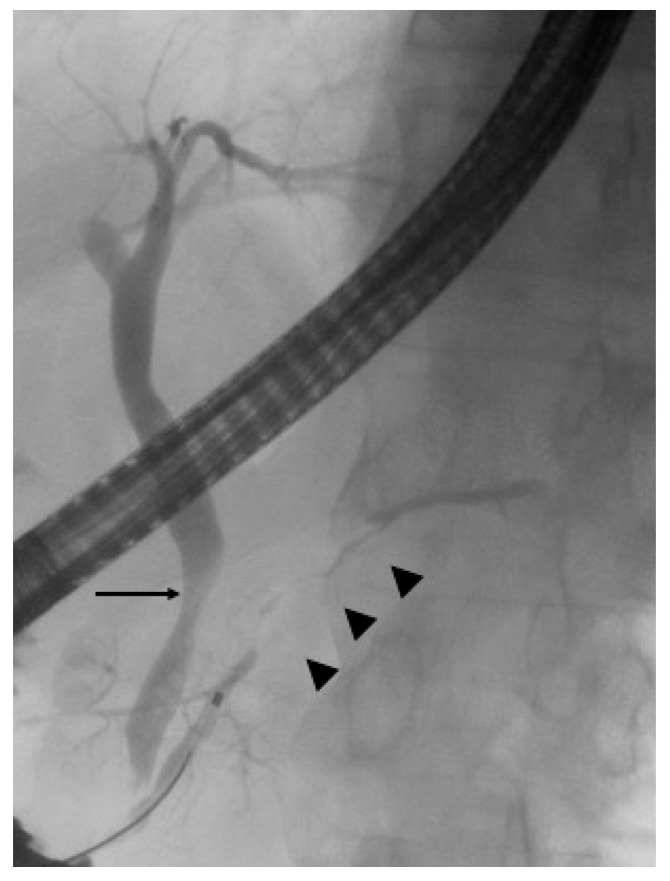
Endoscopic retrograde pancreatography revealed diffuse irregular narrowing of the main pancreatic duct (arrowhead) and stenosis of the lower bile duct (arrow).

**Figure 2 diagnostics-10-01005-f002:**
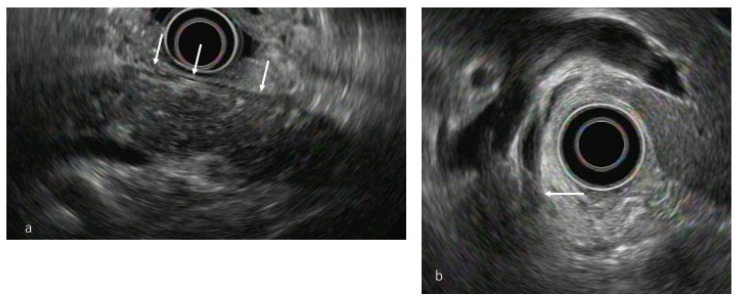
Endoscopic ultrasonography (EUS). (**a**) Conventional EUS findings showed diffuse pancreatic enlargement with a heterogeneous hypoechoic pattern (arrow). (**b**) EUS revealed wall thickening in the distal bile duct (arrow).

**Figure 3 diagnostics-10-01005-f003:**
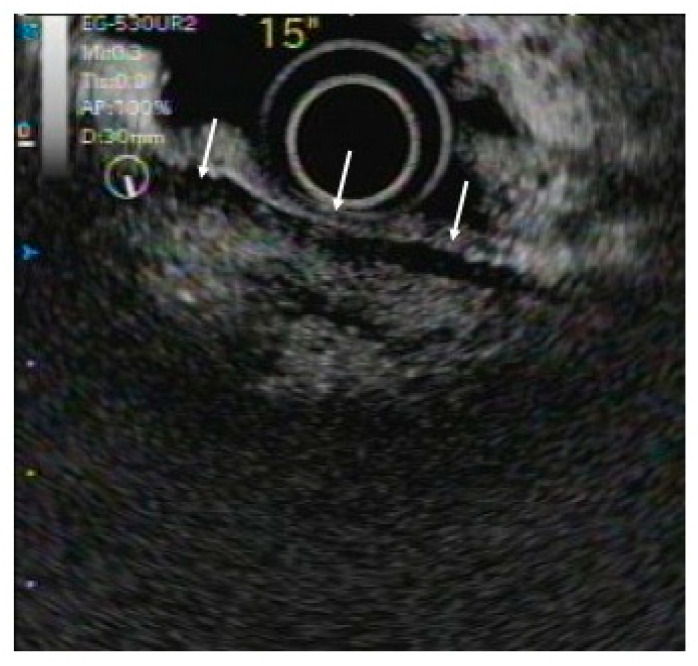
Contrast-enhanced harmonic EUS (CEH-EUS). CEH-EUS findings revealed hypervascular pancreatic enlargement surrounded by hypovascular lesions (capsule-like rim) (arrow).

**Figure 4 diagnostics-10-01005-f004:**
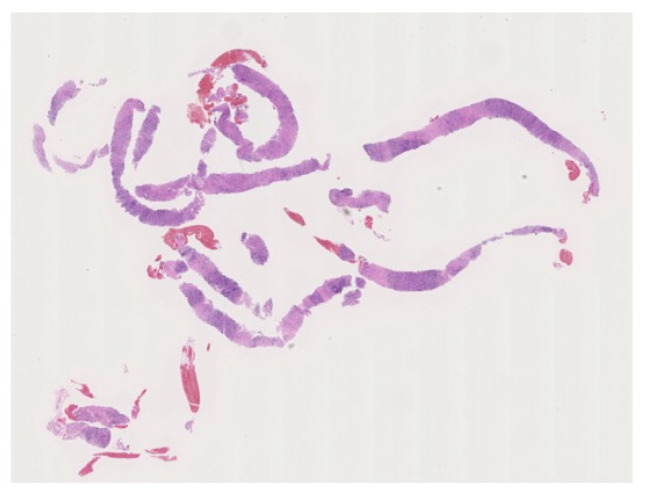
Macroscopic findings of specimens obtained using EUS-guided fine needle biopsy (EUS-FNB) with a 22G needle yielded adequate specimens for histopathological diagnosis.

**Figure 5 diagnostics-10-01005-f005:**
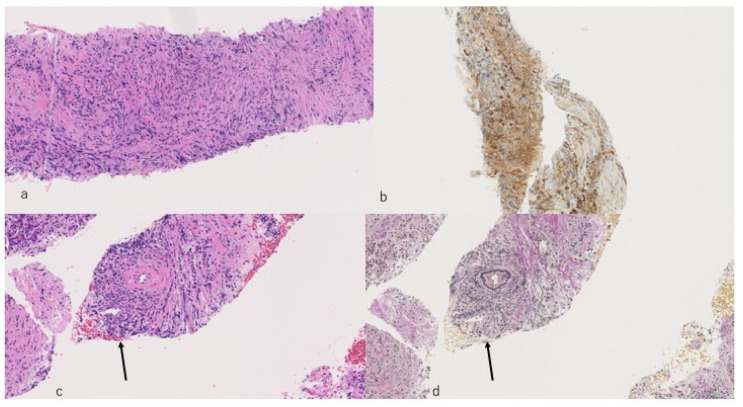
Histological findings obtained using an EUS-FNB needle. (**a**): Lymphoid cell infiltration and storiform fibrosis. (**b**): IgG4 positive plasma cell. (**c**): Obliterative phlebitis (HE) (arrow) (**d**): Obliterative phlebitis (Elastica van Gieson (EVG) staining) (arrow); EVG staining clearly shows findings of obliterative phlebitis.

**Figure 6 diagnostics-10-01005-f006:**
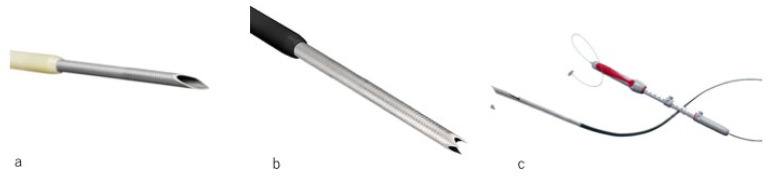
EUS-guided fine needle aspiration (EUS-FNA) and EUS-FNB needles. (**a**) EUS-FNA needle (Lancet needle (Expect Endoscopic Ultrasound Aspiration Needle; Boston Scientific Corp, Natick, MA, USA)). (**b**) EUS-FNB needle (Franseen-like needle (Acquire; Boston Scientific Corp, Natick, MA, USA)). (**c**) EUS-FNB needle (Side-fenestrated needles (ProCore; Cook Medical, Bloomington, IN, USA)).

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
