# Peer review of "The Diagnosis of Autoimmune Pancreatitis Using Endoscopic Ultrasonography"

_diagnostics, 2020, doi:10.3390/diagnostics10121005_

Round 1

Reviewer 1 Report

The paper is a comprehensive review for the diagnosis of AIP using EUS. Despite the title, other available imaging modallities have been discussed, namely ERCP IDUS, both for AIP diagnosis and for IgG4-related cholangitis (IgG4-RC) involving common bile duct. The paper is well write, with complete references, but I suggest some modification to improve the quality of the paper.

  1. The Authors introduced ERCP as a "an essential modality for the diagnosis of AIP and is an essential procedure for intraductal ultrasonography." I agree with Authors, but IDUS is not reported as standard imaging modality for diagnosis of IgG4-RC, and it is not included in ICDC. Furthermore, ICDC suggest also MRI with MRCP sequencies instead ERCP, because in western countries this endoscopic procedure can be used only for therapeutic procedure. MRI/MRCP allow to evaluate not onlu ductal morphology, but also pancreatic parenchima, not investigated by ERCP, and extra-pancreatic involvement of the disease (biliary tree, renal, aorta, retroperitoneum). I Think that all these aspects need to be discussed.
  2. An important meta-analysis is not included in the paper (Contrast-enhanced EUS for differential diagnosis of pancreatic mass
    lesions: a meta-analysis. Gastrointest Endosc 2012;76:301-9), were sensitivity and specificity of CE-EUS to differentiate pancreatic adenocarcinoma vs other pancreatic masses was 94% and 89%, respectively.
  3. In clinical practice, one problem in EUS guided biopsies was the size of the mass. Recently, a paper dimonstrated that mass size affects the accuracy of EUS-FNA of solid pancreatic lesions (Diagnostic yield of EUS-FNA of small ≤15 mm solid pancreatic lesions using a 25-gauge needle. Hep Panc Dis Int, 2018:17:70–74). Can the Authors discuss also this point.
  4. The Authors should stress in the paper that EUS guided FNAC o FNAB is not useful if AIP diagnosis has been achieved with ICDC criteria. A paper by the first Author of this paper (ref #55) reported that histological diagnosis of AIP cannot be reached by histology 11 out of 25 patients (44%) with definitive diagnosis by ICDC, and further 5 patients (20%) had only level 2 ICDC histology (not definitive alone).

Author Response

November 18, 2020

Prof. Dr. Andreas Klaer

Editor-in-Chief, Diagnostics

Re: Manuscript ID  diagnostics-999941

Dear Dr. Klaer:

We are grateful for your kind and helpful comments. Please find our revised manuscript entitled “The Diagnosis of Autoimmune Pancreatitis using Endoscopic Ultrasonography” which we would like to submit for publication in Diagnostics.

We have carefully checked the reviewers’ helpful and constructive comments, and the substantial issues raised by reviewers have been addressed below.

To Reviewer 1 (REV1):

  1. The Authors introduced ERCP as a "an essential modality for the diagnosis of AIP and is an essential procedure for intraductal ultrasonography."I agree with Authors, but IDUS is not reported as standard imaging modality for diagnosis of IgG4-RC, and it is not included in ICDC. Furthermore, ICDC suggest also MRI with MRCP sequencies instead ERCP, because in western countries this endoscopic procedure can be used only for therapeutic procedure. MRI/MRCP allow to evaluate not only ductal morphology, but also pancreatic parenchyma, not investigated by ERCP, and extra-pancreatic involvement of the disease (biliary tree, renal, aorta, retroperitoneum). I think that all these aspects need to be discussed.

- Thank you for this comment. As suggested by the REV1, we have added the description of MRI/MRCP to diagnose autoimmune pancreatitis on page 3, line 95-99 and the reference 26-28.

  1. An important meta-analysis is not included in the paper (Contrast-enhanced EUS for differential diagnosis of pancreatic mass lesions: a meta-analysis. Gastrointest Endosc 2012;76:301-9), were sensitivity and specificity of CE-EUS to differentiate pancreatic adenocarcinoma vs other pancreatic masses was 94% and 89%, respectively.

- As suggested by the REV1, we have added a description of this meta-analysis on page 9, line 292-296 and the reference 79.

  1. In clinical practice, one problem in EUS guided biopsies was the size of the mass. Recently, a paper demonstrated that mass size affects the accuracy of EUS-FNA of solid pancreatic lesions (Diagnostic yield of EUS-FNA of small ≤15 mm solid pancreatic lesions using a 25-gauge needle. Hep Panc Dis Int, 2018:17:70–74). Can the Authors discuss also this point.

-As suggested by the REV1, we have added the description about the size of localized AIP on page 9, line 289-291 and the reference 78.

  1. The Authors should stress in the paper that EUS guided FNAC or FNAB is not useful if AIP diagnosis has been achieved with ICDC criteria. A paper by the first Author of this paper (ref #55) reported that histological diagnosis of AIP cannot be reached by histology 11 out of 25 patients (44%) with definitive diagnosis by ICDC, and further 5 patients (20%) had only level 2 ICDC histology (not definitive alone).

-Thank you for this comment. We apologize for the ambiguous description. The diagnostic ability regarding our paper was low based on reference 58. However, the diagnostic ability of cases to obtain sufficient histological specimens was very high. We changed the description to understand more easily as suggested by REV1 on page 6, line 205-208.

We hope that our paper is now acceptable for publication in Diagnostics.

Very sincerely,

Atsushi Kanno, MD, PhD

Department of Medicine, Division of Gastroenterology, Jichi Medical University, Shimotsuke, Japan

3311-1 Yakushiji, Shimotsuke, Tochigi, 329-0498, Japan

Tel: 81-285-58-7348, Fax: 81-285-44-8297

Reviewer 2 Report

  1. The manuscript described in detail for the diagnosis of AIP using endoscopic ultrasonography. However, in some sections such as the FNA procedures concerning about the techniques and processing of the pathological specimens, the description rendered the subject of AIP diagnosis less likely the main point.
  2. There seemed to have some language problems and also spelling errors, it is better rechecked by an English native personnel to review again..
  3. The Fig 5 item labels: a. c. c. d. e. should be a. b. c. d. e.
  4. For the Fig 3 legend: CEH-EUS findings revealed hypervascular pancreatic enlargement surrounding hypovascular lesions (capsule-like rim) (arrow).--> should it be “CEH-EUS findings revealed hypervascular pancreatic enlargement surrounded by hypovascular lesions (capsule-like rim)”?

Author Response

November 20, 2020

Prof. Dr. Andreas Klaer

Editor-in-Chief, Diagnostics

Re: Manuscript ID  diagnostics-999941

Dear Dr. Klaer:

We are grateful for your kind and helpful comments. Please find our revised manuscript entitled “The Diagnosis of Autoimmune Pancreatitis using Endoscopic Ultrasonography” which we would like to submit for publication in Diagnostics.

We have carefully checked the reviewers’ helpful and constructive comments, and the substantial issues raised by reviewers have been addressed below.

To Reviewer 2 (REV2):

1.The manuscript described in detail for the diagnosis of AIP using endoscopic ultrasonography. However, in some sections such as the FNA procedures concerning about the techniques and processing of the pathological specimens, the description rendered the subject of AIP diagnosis less likely the main point.

 -Thank you for this comment. We apologize for the ambiguous description. Since obtaining sufficient histological material by EUS-FNA is very important to diagnose AIP correctly, we described the FNA procedures. We added a description concerning FNA techniques on page 8, lines 233-234.     

2.There seemed to have some language problems and also spelling errors, it is better rechecked by an English native personnel to review again.

- As suggested by the REV2, English language usage has been revised thoroughly by a native English speaking scientist and author.

3.The Fig 5 item labels: a. c. c. d. e. should be a. b. c. d. e.

-We apologize for the careless mistake. We re-labeled Fig 5 item as a. b. c. d. on page 20, lines 10-13.

4.For the Fig 3 legend: CEH-EUS findings revealed hypervascular pancreatic enlargement surrounding hypovascular lesions (capsule-like rim) (arrow).--> should it be “CEH-EUS findings revealed hypervascular pancreatic enlargement surrounded by hypovascular lesions (capsule-like rim)”?

-As suggested by REV1, we have changed the description of the figure legend on page 7, line 222.

We hope that our paper is now acceptable for publication in Diagnostics.

Very sincerely,

Atsushi Kanno, MD, PhD

Department of Medicine, Division of Gastroenterology, Jichi Medical University, Shimotsuke, Japan

3311-1 Yakushiji, Shimotsuke, Tochigi, 329-0498, Japan

Tel: 81-285-58-7348, Fax: 81-285-44-8297
